# Modeling the impact of xenointoxication in dogs to halt *Trypanosoma cruzi* transmission

**Jennifer L. Rokhsar**[1,2,3☯], **Brinkley Raynor**[4☯], **Justin Sheen**[4,5], **Neal D. Goldstein**[2], **Michael Z. Levy**[4,6], **Ricardo Castillo-Neyra**[4,6]*

**1** Faculty of Health and Medical Sciences, University of Surrey, Guildford, United Kingdom, **2** Department of Epidemiology and Biostatistics, Dornsife School of Public Health, Drexel University, Philadelphia, Pennsylvania, United States of America, **3** ORISE Fellow, Emerging Leaders in Data Science and Technologies Program Fellowship, National Institute of Allergy and Infectious Diseases (NIAID), NIH, United States of America, **4** Department of Biostatistics, Epidemiology, and Informatics, Perelman School of Medicine, University of Pennsylvania, Philadelphia, Pennsylvania, United States of America, **5** Department of Ecology and Evolutionary Biology, Princeton University, Princeton, New Jersey, United States of America, **6** One Health Unit, School of Public Health and Administration, Universidad Peruana Cayetano Heredia, Lima, Peru

☯ These authors contributed equally to this work.

\* cricardo@upenn.edu

**Data Availability Statement:** All relevant data are within the manuscript and its Supporting Information files.

## Abstract

### Background

Chagas disease, a vector-borne parasitic disease caused by *Trypanosoma cruzi*, affects millions in the Americas. Dogs are important reservoirs of the parasite. Under laboratory conditions, canine treatment with the systemic insecticide fluralaner demonstrated efficacy in killing *Triatoma infestans* and *T. brasiliensis*, *T. cruzi* vectors, when they feed on dogs. This form of pest control is called xenointoxication. However, *T. cruzi* can also be transmitted orally when mammals ingest infected bugs, so there is potential for dogs to become infected upon consuming infected bugs killed by the treatment. Xenointoxication thereby has two contrasting effects on dogs: decreasing the number of insects feeding on the dogs but increasing opportunities for exposure to *T. cruzi* via oral transmission to dogs ingesting infected insects.

### Objective

Examine the potential for increased infection rates of *T. cruzi* in dogs following xenointoxication.

### Design/Methods

We built a deterministic mathematical model, based on the Ross-MacDonald malaria model, to investigate the net effect of fluralaner treatment on the prevalence of *T. cruzi* infection in dogs in different epidemiologic scenarios. We drew upon published data on the change in percentage of bugs killed that fed on treated dogs over days post treatment. Parameters were adjusted to mimic three scenarios of *T. cruzi* transmission: high and low disease prevalence and domestic vectors, and low disease prevalence and sylvatic vectors.

**Funding:** BHR was supported through an NIH grant 5-T32-AI-070077-14. RCN was supported by NIH-NIAID grant 1K01AI139284, https://www.nih.gov/. The funders had no role in study design, data collection and analysis, decision to publish, or preparation of the manuscript.

**Competing interests:** The authors have declared that no competing interests exist.

## Results

In regions with high endemic disease prevalence in dogs and domestic vectors, prevalence of infected dogs initially increases but subsequently declines before eventually rising back to the initial equilibrium following one fluralaner treatment. In regions of low prevalence and domestic or sylvatic vectors, however, treatment seems to be detrimental. In these regions our models suggest a potential for a rise in dog prevalence, due to oral transmission from dead infected bugs.

## Conclusion

Xenointoxication could be a beneficial and novel One Health intervention in regions with high prevalence of *T. cruzi* and domestic vectors. In regions with low prevalence and domestic or sylvatic vectors, there is potential harm. Field trials should be carefully designed to closely follow treated dogs and include early stopping rules if incidence among treated dogs exceeds that of controls.

## Author summary

Chagas disease, caused by the parasite *Trypanosoma cruzi*, is transmitted via triatomine insect vectors. In Latin America, dogs are a common feeding source for triatomine vectors and subsequently an important reservoir of *T. cruzi*. One proposed intervention to reduce *T. cruzi* transmission is xenointoxication: treating dogs with oral insecticide to kill triatomine vectors in order to decrease overall *T. cruzi* transmission. Fluralaner, commonly administered to prevent ectoparasites such as fleas and ticks, is effective under laboratory conditions against the triatomine vectors. One concern with fluralaner treatment is that rapid death of the insect vectors may make the insects more available to oral ingestion by dogs; a more effective transmission pathway than stercorarian, the usual route for *T. cruzi* transmission. Using a mathematical model, we explored 3 different epidemiologic scenarios: high prevalence endemic disease within a domestic *T. cruzi* cycle, low prevalence endemic disease within a domestic *T. cruzi* cycle, and low prevalence endemic disease within a semi-sylvatic *T. cruzi* cycle. We found a range of beneficial to detrimental effects of fluralaner xenointoxication depending on the epidemiologic scenario. Our results suggest that careful field trials should be designed and carried out before wide scale implementation of fluralaner xenointoxication to reduce *T. cruzi* transmission.

## Introduction

Chagas disease, a vector-borne neglected tropical disease, affects millions of people in Latin America [1,2]. The disease, caused by the protozoan *Trypanosoma cruzi*, has no specific treatment, and there are no vaccines available for a large-scale public health intervention; therefore, strategies to control and eliminate Chagas disease have targeted the insect vectors, triatomine bugs [3]. Importantly, transmission of *T. cruzi* occurs in two main cycles [4]. The sylvatic cycle involves small wild mammals acting as animal reservoirs and sylvatic bugs bringing the parasite into households, infecting humans and domiciliary mammals. The domestic cycle involves the colonization of household structures by triatomine bugs and the transmission of the parasite to and from humans and domiciliary mammals. However, there are regions where these

two cycles overlap, and some authors recognize the existence of a peri-urban cycle [4]. The presence of multiple wild mammalian reservoirs makes elimination virtually impossible in the sylvatic cycle [5], but within the domestic cycle dogs are much more accessible reservoirs than wild animals to target with One Health interventions for elimination of *T. cruzi* transmission and reduction of Chagas disease in humans [6].

Triatomine bugs acquire infection through blood meals from mammals that contain infective forms of *T. cruzi* in their bloodstream [7]. By contrast hosts can become infected with *T. cruzi* through several avenues, including congenital and oral, but the most common and important is vector-borne transmission [8]. Oral transmission through predation of infected vectors is thought to be the most frequent mechanism of infection among hosts in the sylvatic cycle [9–14] and many people have been infected orally in focalized outbreaks in Latin America [9,15]. The probability of transmission due to oral vector ingestion is estimated to be about 1000 times greater than vectorial transmission [16,17] and *T. cruzi* parasites in feces outside of the bug are viable (infectious) for up to 48 hours (e.g., in fruit juices) [17]. Dogs, important reservoirs for Chagas disease in the domestic cycle [18–21], can occasionally get infected through the oral route [22].

Dogs are key reservoirs in the urban and sylvatic cycles of *T. cruzi* because they are very common within households in Latin America [6,23], they have longer lifespans compared to other important animal reservoirs such as guinea pigs [24,25], they act as bridges between both cycles [25], and dogs tend to have high infection rates, are very infectious, and have high rates of contact with both vectors and humans [20]. In Latin America, reports on canine seroprevalence in areas where natural infection occurs concentrate between 8–28%, with extremes of 1.5% and 83.3% [26–28]. Infected dogs are also 100 times more likely to infect susceptible triatomes than infected adult humans and 12 times more infectious than infected children [29]. In Brazil, *Triatoma brasiliensis*, one of the primary vectors of *T. cruzi*, overlaps geographically with areas where dogs are important reservoirs of the disease [30,31]. In addition, *Triatoma infestans*, one of the primary insect vectors of *T. cruzi* in South America, shows consistent preference for dogs over other domestic animals [29]. The strong preference triatomine vectors have for dogs can be exploited via xenointoxication–a targeted vector control strategy where insecticides are administered to peri-domestic and domestic animals (e.g., dogs) to suppress insect infestations. For instance, by targeting dogs with topical insecticides (or insecticide impregnated collars), dogs are effectively turned into baited lethal traps [6].

Interventions on the dog population to eliminate *T. cruzi* transmission have been evaluated for decades. Mathematical models of Chagas disease have shown that removal of infected dogs from a household containing infected people could stop disease transmission (excluding reintroduction) [32], but culling the dog population would be, at the very least, socially unacceptable and hypothesized to have inconclusive results [6]. Recent experimentation treating dogs with oral or topically applied insecticides showed promising efficacy at killing triatomines [33–35]; in particular, fluralaner, a relatively new isoxazoline oral insecticide commonly used to prevent tick and flea infestations, proved especially effective in killing bugs when they fed on dogs under laboratory conditions [33] and is being considered for Chagas control programs [36]. As the unit cost for indoor residual insecticide treatment in a rural house is quite high [37–40] and can be met with low levels of community participation [41–44], treatment of canine reservoirs with insecticide could prove a useful alternative or complementary strategy to reduce *T. cruzi* infection in people. Additionally, due to the scarcity of insecticide for public health usage [45], treatment of canines with a safe, long-lasting, effective insecticide such as fluralaner potentially could prove a valuable tool in the face of pyrethroid shortage [6,33]. However, given that *T. cruzi* can be easily transmitted orally through the ingestion of triatomines [9,13,15,46–49], there is potential for a counterproductive effect: dogs could consume

the infected bugs killed by the treatment [27,35,50,51], increasing infection rates in the dog population.

Xenointoxication as an intervention for Chagas disease could have unexpected consequences. The use of fluralaner could potentially reduce *T. cruzi* transmission by reducing the number of infectious bugs; however, it is also possible that the use of fluralaner could increase *T. cruzi* transmission by making infectious bugs killed by treatment more orally available to dogs. In this study, we developed a deterministic model of *T. cruzi* transmission dynamics that accounts for both vector-borne transmission and transmission via ingestion of *T. cruzi*-infected triatomines in dog populations. We used the model to investigate the effects the intervention will have on the prevalence of infections among insects and dogs under a variety of epidemiologic scenarios.

## Results

### Pretreatment model

In regions affected by the domestic cycle and high prevalence of disease, prior to the administration of fluralaner treatment, equilibrium prevalence for dogs was 53.68% and for bugs it was 54.48% at approximately 10,000 days (27.4 years) (S1 Fig). In regions affected by the sylvatic cycle and low prevalence of disease, prior to the administration of treatment, equilibrium prevalence for dogs was 23.64% and for bugs it was 38.81% at approximately 20,000 days (54.8 years) (S1 Fig). As the ratio of bugs to dogs in the population goes from 5–100, population dynamics switch from one where the proportion of infected bugs exceeds the proportion of infected dogs to the reverse. Pretreatment, the parameter with the largest impact on transmission dynamics is dogs' lifespan; in populations where dogs live for $\geq 3$ years, there are higher rates of overall infection for both bug and dog populations. These parameters and the potential xenointoxication interventions (e.g., number of treatments) can be modified in our interactive visualization application found at https://jrokh.shinyapps.io/NewExternalBugs/.

### Treatment model: Domestic Vectors

We explored several different aspects of treatment, including the frequency of treatment and the length between treatments. A single treatment of fluralaner after population equilibrium resulted in a sharp decline of the proportion of infected bugs and a simultaneous increase in the proportion of infected dogs immediately after treatment (Fig 1). The rise in the proportion of infected dogs is followed by a gradual decline and a rise back to equilibrium levels. The sharp decline in the proportion of infected bugs also rises back to equilibrium levels. The percentage of bugs consumed by dogs will be a function of both individual dog behavior and accessibility, i.e., bugs die in a location that is accessible to the dog; therefore, we varied the percentage of dead bugs consumed by dogs. In this simulation, we assumed that dogs consumed 80% of bugs killed with fluralaner treatment; in simulations with this parameter set to 20% and 50%, trends remained the same (S2 Fig). As could be expected, if the dogs consumed a greater number of the bugs, the initial rise in the proportion of infected dogs is greater, followed by a shallower decline in the days post-treatment (DPT).

We examined the effects of administering canine fluralaner treatment once a year for 4–6 years (Fig 2A and 2B). Similar to the effect of single treatment with fluralaner, immediately following administration, the proportion of infected dogs rises followed by a gradual decline. Also, at each successive treatment, there is a corresponding rise in the proportion of infected dogs; however, these peaks remain less than pretreatment equilibrium prevalence. Likewise, each treatment corresponds to a sharp decline in the proportion of infected bugs; as the treatment effect wears off, the proportion of infected bugs rises more rapidly than the infected

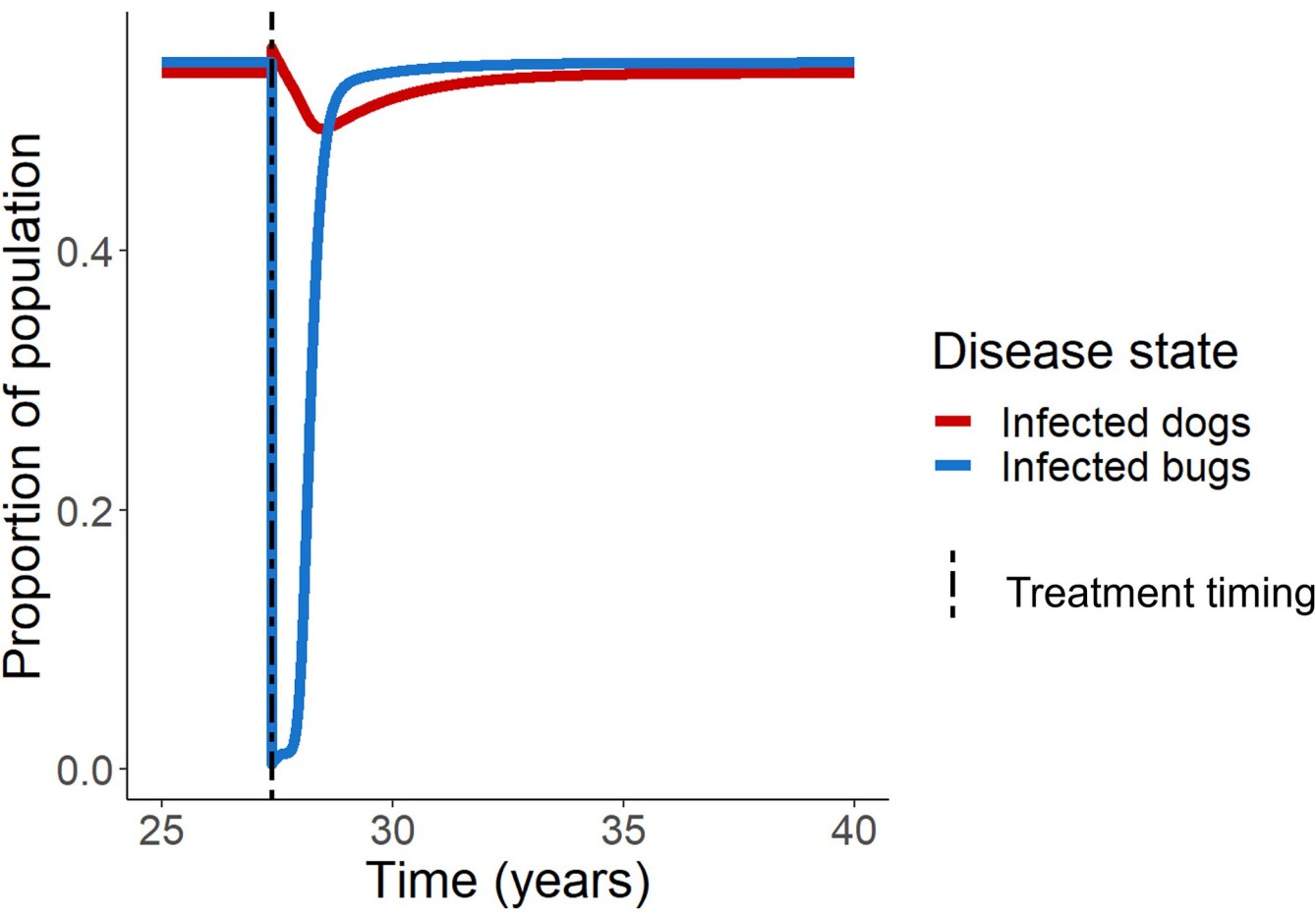

**Fig 1. Single fluralaner treatment in a high prevalence region.** Proportions of dogs and bugs infected with *T. cruzi* after single administration of fluralaner treatment at equilibrium (27.4 years) in a region of high prevalence of endemic disease and domestic vectors was simulated.

dogs, but infection levels still remain less than equilibrium prevalence. Treating every year prevents the infection prevalence in both dogs and bugs to reach prior equilibrium levels; the effect of successive yearly treatment allows for a "stair step" effect, where each peak in dog infection prevalence at treatment administration is smaller than the peak prior. We also explored setting the triatomine birthrate to zero, allowing the triatomine population to crash after xenointoxication treatment. As expected, we found that with no vectors to transmit *T. cruzi*, the proportion of infected dogs declines within years.

Manufacturer's instructions call for oral fluralaner to be given to dogs once every 12 weeks (approximately 90 days) [52]. When fluralaner is given according to this frequency (Fig 2C and 2D), we observe a similar "stair step" effect; there is an initial spike in dog infections, but in subsequent treatments these peaks are smoothed out; even after treatment is stopped, the proportion of infected dogs continues to trend downwards for a period of time before the infection levels begin to climb back towards pretreatment equilibrium levels. Giving treatments at this frequency also suppressed the infected bug proportion from rising between treatments. Levels of infection in the bug population remain low for a period of time following the last treatment before returning back to pretreatment levels. In these simulations, we assumed that dogs consumed 80% of bugs killed with fluralaner treatment; in simulations with this parameter set to 20% and 50%, trends remained the same (S3 Fig).

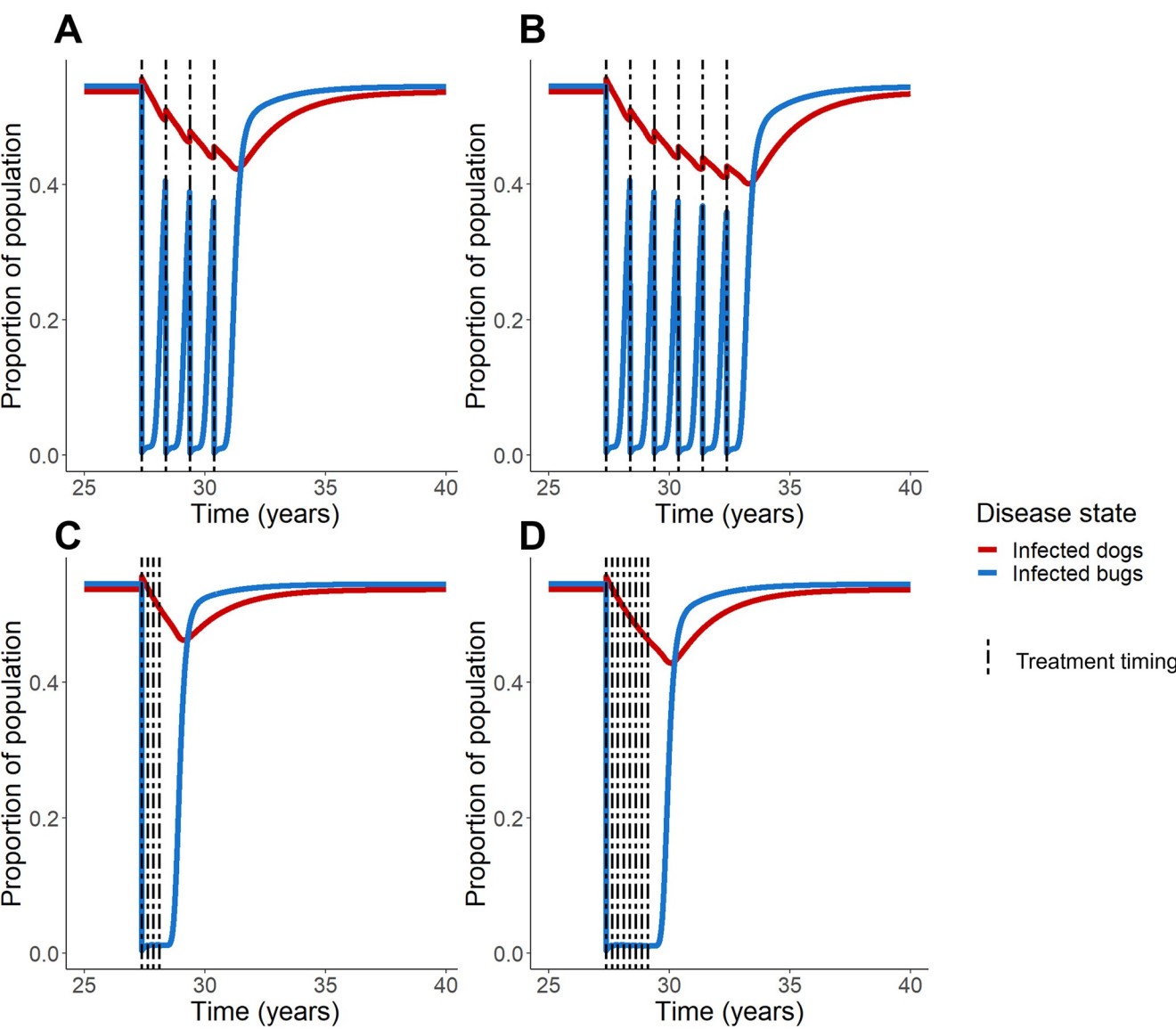

**Fig 2. Multiple fluralaner treatments in a high prevalence region.** Treatment scenarios were simulated for equilibrium populations of bugs and dogs in a region of high prevalence of endemic disease and domestic vectors. Annual administration of fluralaner for both 4 years (A) and 6 years (B) was simulated, as well as administration every 90 days (veterinary recommendation) for one year (C) and for two years (D).

We examined the effects of treatment on areas with domestic vectors and a low prevalence of disease (Fig 3). In regions with a low prevalence of disease ($m = 15$) and dogs average lifespan = 3 years, fluralaner treatment is marked by the initial increase in prevalence of infected dogs (at time of treatment), but -unlike regions with a high prevalence of disease- the infection peak does not gradually decline; rather, it forms an elevated plateau followed by a gradual decline back to equilibrium infection levels. In regions of low disease prevalence ($m = 7$) and dogs with longer lifespans (average lifespan = 6 years), the initial spike in dog infection prevalence continues to rise for a period of time; with each successive treatment, the proportion of infected dogs rises higher than the peak prior (Fig 3B). When percentages of dead bugs consumed by dogs is lower (at 20% and 50% instead of 80%), the level of infected dogs decline with fluralaner treatment when dogs have a 3-year life span (S4A–S4C Fig). When dogs have a

          

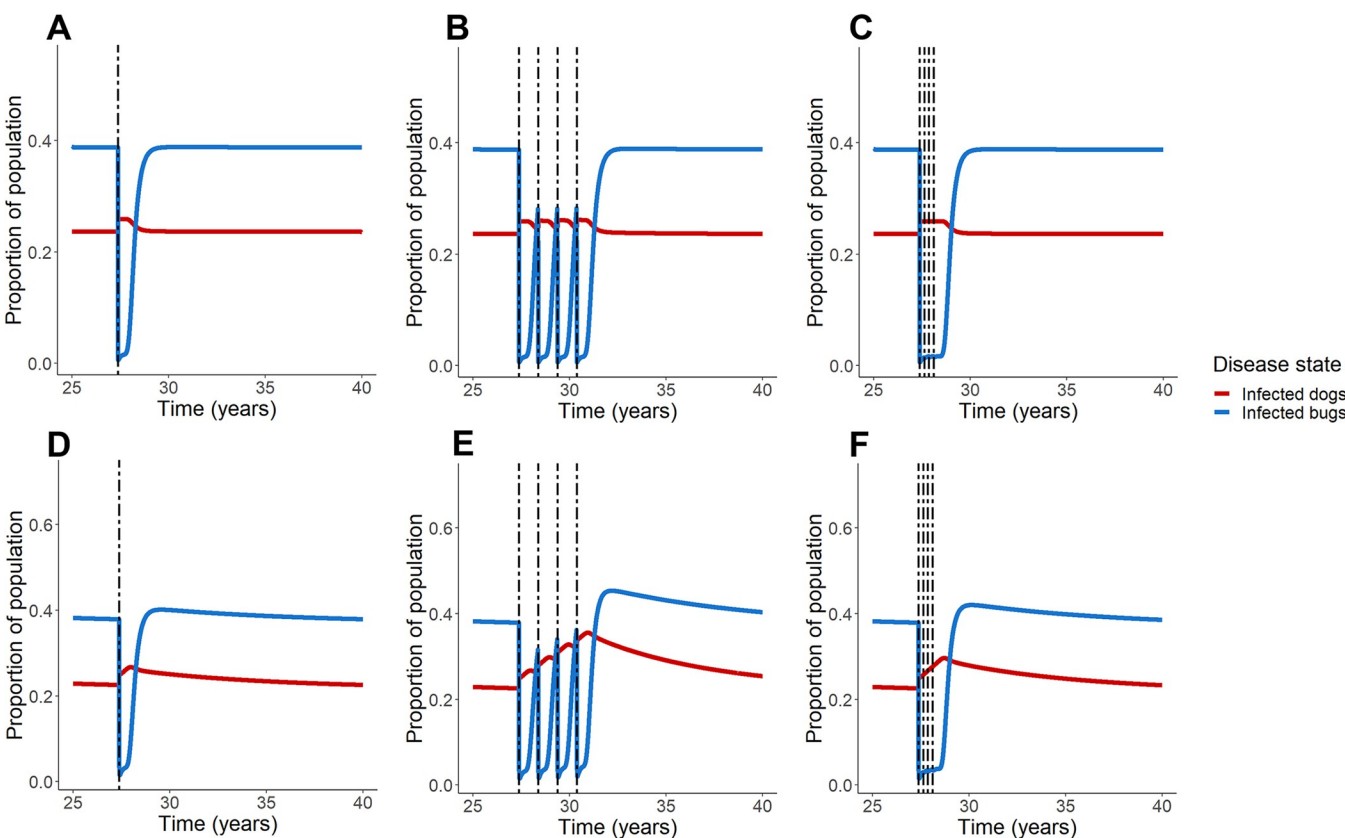

**Fig 3. Fluralaner treatment schemes in low prevalence regions.** Simulations were conducted to explore the effect of fluralaner treatment of regions of low prevalence of endemic disease and domestic vectors in equilibrium; we explored a range of dog average lifespan from 3 years (A-C) to 6 years (D-F). Treatment scenarios include one time treatment (A, D), annual treatment for 4 years (B, E), and treatment every 90 days for 1 year (C, F).

6-year lifespan, infected dog numbers trend up after fluralaner treatment assuming a percentage of 50% as well as 80% but trend down with 20% (S4D–S4F Fig).

## Treatment model: Semi-sylvatic vectors

We simulated regions with lower disease prevalence and semi-sylvatic vectors for both the baseline average dog lifespan (Fig 4A–4C) as well as the 6-year life span (Fig 4D–4F). Similar to the prior models with low disease prevalence, administration of fluralaner leads to a rise in dog infection prevalence, which increases with successive treatments. The effect is particularly apparent where dogs have longer lifespans (Fig 4D–4F); although bug infection experiences a sharp decline upon treatment administration, with repeated treatments, the bug infection prevalence rebounds to levels above the pretreatment equilibrium values. The semi-sylvatic low-prevalence model is sensitive to the proportion of bugs eaten with trends being similar to those of the non-semi-sylvatic cycle low-prevalence model (S5 Fig).

## Discussion

Our models indicate that in regions with high disease prevalence and domestic vectors treatment of dogs with fluralaner could provide an effective complementary community-level treatment of *T. infestans* infestations, similar to what lab experiments suggest [34]. In regions with high prevalence of household infestations, even if dogs were to consume large numbers of *T*.

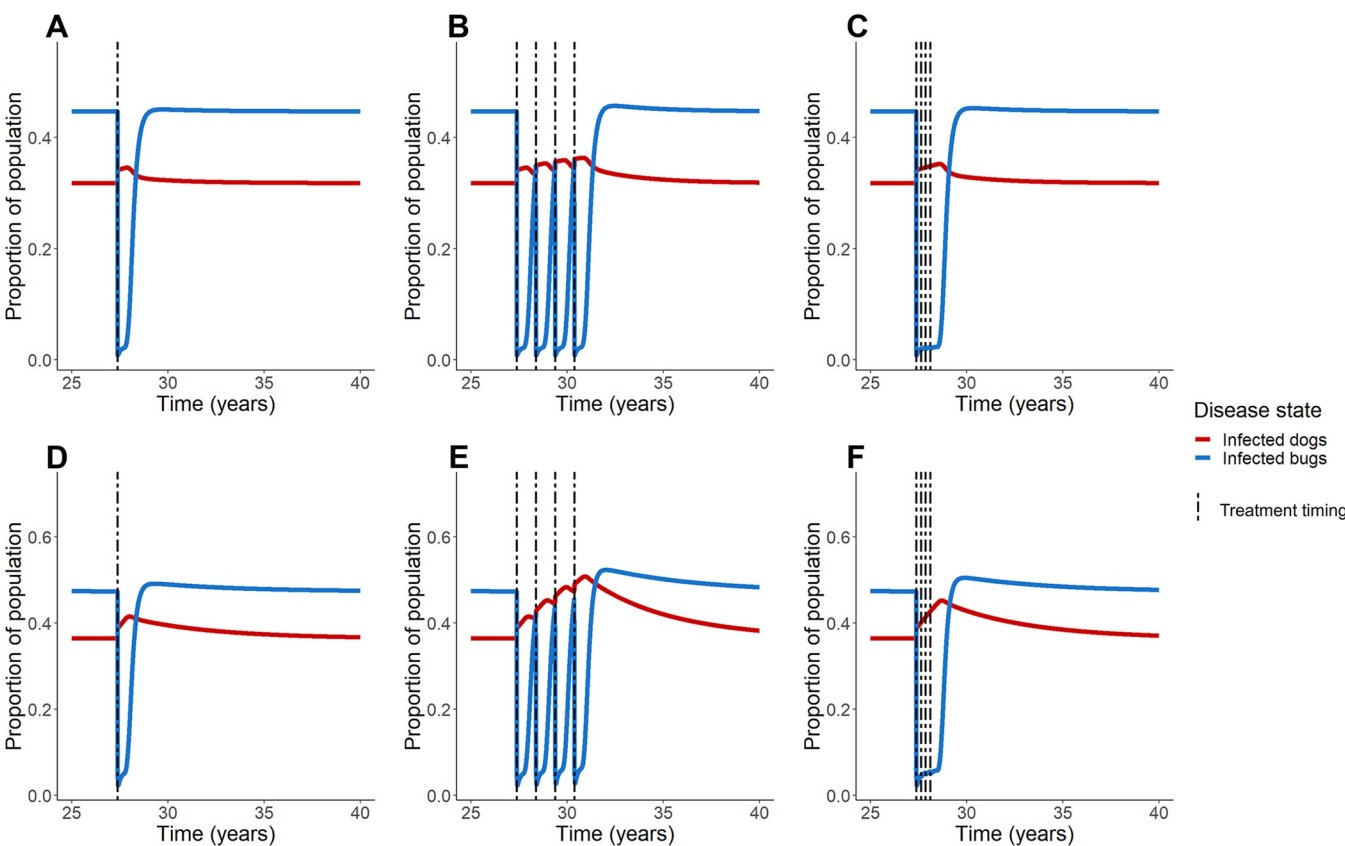

**Fig 4. Fluralaner treatment schemes in low prevalence regions with semi-sylvatic transmission.** Simulations were conducted to explore the effect of fluralaner treatment of regions of low prevalence of endemic disease and domestic vectors as well as semi-sylvatic vectors in equilibrium; we explored a range of dog average lifespan from 3 years (A-C) to 6 years (D-F). Treatment scenarios include one time treatment (A, D), annual treatment for 4 years (B, E), and treatment every 90 days for 1 year (C, F).

*cruzi*-infected bugs, our models suggest that levels of canine infection would drop below pretreatment levels following the initial rise due to oral consumption. Even more promising, treatment appears to be beneficial if given at yearly intervals, which would be more cost-effective, and likely have higher community participation rates, than treating every 3 months. As a point of comparison, the price of fluralaner for a medium size dog in Peru is 22.20 USD [53] and the minimum wage in the same country is 275 USD/month [54]. Our model ignored seasonality; it could be possible to time yearly treatments leading to a slower resurgence of the vector population the following year, similar to timing of spraying campaigns [55,56]. The findings of these simulations are supported by a placebo controlled before-and-after efficacy trial of fluralaner administration to dogs in Chaco Province, Argentina (a region with high prevalence of domestic vectors/household infestation); the authors demonstrated that site infestation and domicile bug abundance plummeted over the months posttreatment [57].

In contrast, our findings suggest that in regions with low disease prevalence and domestic or sylvatic bug populations, especially in regions where dogs have longer lifespans, careful attention needs to be given to the potential of unintended consequences of xenointoxication on *T. cruzi* transmission. In these regions, when dogs are able to consume a large percentage of the bugs (50% or more based on our sensitivity analysis for 6-year life-spans) our models suggest that infection levels in dogs (and in some situations infection in bugs), end up higher than pretreatments levels. As the rise in dog infection prevalence occurred either at treatment

administration or shortly thereafter, when bug infection prevalence is very low, we can say that this expected increase is due to canine consumption of bugs killed by treatment. Whether a dog would consume a bug killed by treatment would depend on 1) dog behavior and 2) whether the bugs would be able to conceal themselves prior to succumbing to the treatment.

It was reported that nearly all the bugs that fed on treated dogs between 4–60 DPT died within 24 hours of exposure [33], and we assumed that dogs consumed 80% of bugs killed by treatment and that consumption happened immediately upon death, but we report results from 20% and 50% consumption in the supplement. The consequence of relaxing this assumption, and having the dogs consume a smaller percentage, would reduce the risk of oral infection, in some cases making treatment beneficial in regions with lower disease prevalence and domestic vectors. Detailed data on the time distribution from bugs feeding on dogs to death would improve estimates for the percentage of bugs that dogs would be able to consume. Our findings warrant further lab experiments and small field trials before launching large xenointoxation-based elimination programs. Before utilizing fluralaner in regions with low disease prevalence and domestic or sylvatic bug populations, especially in regions where dogs have longer lifespans, lab studies could inform how quickly bugs died within 24-hour post feeding period to refine the estimates of risk of oral transmission post xenointoxication. Better yet, randomized controlled field trials could be designed to closely follow treated dogs, conduct continuous interim analysis, and include early stopping rules if it turns out that treated dogs are becoming infected at rates higher than controls.

Current vector control for *T. infestans* is based on insecticide spray and threatened by the emergence of pyrethroid resistant bugs [58]. Under experimental conditions, fluralaner proved efficacious against both pyrethroid susceptible and resistant 5th-stage nymphs [33]. In fact, between 4–60 DPT, regardless of pyrethroid susceptibility status, almost all bugs were killed after feeding on treated dogs; it would not be until 90–120 DPT that cumulative mortality declined at a greater rate for susceptible bugs than resistant, and these results were found to not be statistically significant [33]. We incorporated the data from Laiño et al. on the percentage of bugs killed after feeding on treated dogs for both the 5th-stage susceptible and resistant nymphs into the Shiny web application, but the difference in model outcome was negligible, regardless of the parameter set.

As fluralaner is a relatively new isoxazoline compound, approved for use in the United States in 2014 (Food and Drug Administration [FDA], 2014), literature review resulted in no information regarding possible fluralaner resistance. Isoxazoline compounds are potent inhibitors of γ-aminobutyric acid (GABA)-gated chloride channels (GABACls) [59]. Previous pharmacological profiles regarding cyclodiene resistance in *Drosophila spp*. demonstrated that resistance to cyclodiene conferred broad cross resistance to compounds blocking GABACls [60]; It has been noted that use of novel chloride channel antagonists as insecticides should be managed carefully in order to prevent the rapid development of field resistance [60]. As fluralaner has shown promise in regards to vector control in regions where *T. infestans* have resistance to pyrethroids [33], careful consideration should go into planning and implementation of community-level canine fluralaner treatment programs to avoid selecting for vectors that develop resistance toward isoxazoline compounds.

There was some uncertainty inherent in several parameter estimates. Our model, describing household *T. cruzi* transmission dynamics, is sensitive to the parameter $m$, the ratio of the number of vectors feeding on any given host; households with a smaller ratio demonstrated unfavorable outcomes with fluralaner treatment when dogs consumed 4 out of 5 of the killed bugs. Yet in small households, populations of domestic animals can be unstable, creating unpredictable fluctuations in this ratio [61]. Likewise, our model only assumed one host, dogs; in a real-world context, the effectiveness of fluralaner treatment on reducing *T. infestans*

infestation would depend in part on the availability of alternative hosts, including humans, chickens and untreated dogs [33]. Experimental studies reported the majority of fed bugs were fully engorged after feeding on fluralaner treated dogs [33,34], making it unlikely that fluralaner has a repellent effect which could divert bug feeding towards humans [6]. Field studies in Argentina suggest the fraction of domestic *T. infestans* with a blood meal on dogs ranging upward of 65%, and that the more bugs fed on dogs the less they fed on humans [62]; it is likely that even with alternative hosts available fluralaner could potentially reduce *T. cruzi* transmission in regions with high disease prevalence and household *T. infestans* infestations.

Chagas disease dynamics are complex and vary much geographically. From our results, it is clear that the impact of fluralaner on halting *T. cruzi* transmission depends on a combination of parasite prevalence, insect abundance, and type of triatomine vectors (domestic vs. sylvatic bugs). We developed a Shiny web application to allow users to alter the transmission and treatment parameters and examine the results according to local conditions. For our models we used the simplifying assumption that dogs have a constant rate of infectiousness and only leave the infected compartment through death. But similar to humans, dogs experience acute and chronic phases of infection [63]; it is during the acute phase that parasitemia is highest. Taking into account varying reports on the duration of parasitemia (Machado et al., 2001) [63,64], the potential for reactivation, and reports of "super-shedders" in other species (guinea pigs) [65], we countered the assumptions of homogeneity and temporal scales of transmission by reducing the probability of transmission between dog and bug from the reported 0.49 [29] to 0.28.

Our model demonstrates the potential for canine fluralaner treatment to reduce *T. cruzi* transmission in regions with high disease prevalence and domestic vectors; fluralaner treatment could be used as a complementary, community-level intervention to reduce *T. infestans* populations in infested households and could be done as infrequently as once a year. On the other hand, in low endemic regions and regions with sylvatic bugs, canine treatment with fluralaner could potentially increase infection prevalence in both dog and bug populations via canine oral consumption of vectors killed by treatment. These simulations, though a simplified version of reality, highlight the need for well-designed studies to investigate the conditions under which fluralaner xenointoxication, a promising One Health intervention, is an effective control strategy against Chagas disease.

## Methods

### Model construction

We conducted a simulation study, using an adaptation of the classic Ross-MacDonald malaria model. Oral predation of triatomines is a well-characterized transmission route of T. cruzi [9–17,22]; through simulation we explored the potential effects of transmission of T. cruzi via ingestion by dogs of triatomines killed by fluralaner treatment. Laboratory evidence suggests that metacyclic trypomastigotes are viable in dead triatomines days after triatomine death [66]; additionally, there are some reports of human infection through contamination of fruit juices with dead triatomines [9,11,15,17]. In our model, we explore scenarios in which we assume that dogs consume dead triatomines and that the dead triatomines ingestion shortly after bug death (within a few days) results in transmission scenarios similar to oral predation. This model included the following simplifying assumptions: the host (dog) population is assumed to be homogenous and constant. The vector (bug) population was also assumed to be homogenous but differed from the classic Ross-MacDonald malaria model [67] as a vector birth rate was incorporated to balance the impact of fluralaner treatment (to avoid having the bug population "crash" shortly after administration of insecticide). For simplicity we parameterized the

**Table 1. Parameter values for modified Ross-MacDonald model simulations.**

| Parameter | Description (unit) | Values (range for sensitivity analysis) | Source |
|---|---|---|---|
| X | proportion of dogs infected | – | – |
| Y | proportion of triatomines infected | – | – |
| a | Expected number of bites on dogs per triatomine | 1/14 [1/7-1/21] | [61] |
| m | Equilibrium triatomine density per dog (triatomines per dog) | 40 [10–100] | Estimated from other species [61] |
| n | Length of the incubation period (days) | 45 [10–60] | [61] |
| g | Daily force of triatomine mortality (1/day) | 0.005 [0.001–0.01] | [69] |
| b | Transmission efficiency from infectious triatomine to susceptible dog via bite (1/ number of bites required for transmission) | 0.00068 [0.0005–0.001] | [70] |
| c | Probability of an infection of an uninfected triatomine by biting an infectious dog | 0.28 [0.10–0.49] | Adapted from: [29] |
| r | Daily force of infectious dog mortality (1/day) | $1/(3*365)$ $[(1/(2*365))-1/(8*365)]$ | Varied in accordance to regional variations |
| p | Maximum proportion of vectors eaten by dogs in a day | 0.8 [0.1–0.99] | Varied to account for difference in individual animal behavior patterns |
| k | Transmission efficiency from infectious triatomine to susceptible dog via oral transmission (proportion of oral infection per infected vector consumed) | 0.1 | [16] |
| R | The maximum birthrate at carrying capacity (day/eggs laid) | 0.09 [0.05–0.11] | [71] |
| K | Carrying capacity of vectors per dog | 40 [high prevalence] 15 [low prevalence, 3-year lifespan] 7 [low prevalence, 6-year lifespan] | Varied with assumed population size |
| z | Proportion of triatomines killed by fluralaner treatment | Time dependent covariate values obtained from log curve | [33] |

bug population based on data on *T. infestans* for the domestic cycle and data on *T. dimidiata* for the sylvatic cycle [68]. We made a number of simplifying assumptions: we ignored vector reproductive senescence and seasonality. We assumed that there was no host recovered class (despite the possibility of both treatment and natural recovery) and grouped hosts in the acute and chronic phases of infection into a single infected class although it is known that hosts are more infectious during the acute phase of infection [64,65]. We further assumed that the only way dogs can leave the infected compartment is through death; to account for cyclic parasitemia, the parameter used in the model for transmission probability from dogs to vectors has been halved what has been used in prior models (see Table 1). Lastly, as *T. infestans* primarily exhibits night-feeding behavior to avoid diurnal predators (Schofield, 1985), we assumed that oral transmission only involves the bugs killed by treatment, i.e., there is no oral transmission prior to fluralaner administration. The implications of changing these assumptions are later discussed.

## Pretreatment model

The model considers a single species of host (dogs) and a single vector, which represents different species in different scenarios. We do not consider more complex situations with multiple vector species. All analyses were carried out in the R software environment [72] using the differential equation solver deSolve [73] and Shiny packages [74]. Red and blue lines in Fig 5 illustrate transmission dynamics among dogs and bugs prior to fluralaner treatment, with *X* representing the proportion of infected dogs and *1-X* the proportion of dogs that are susceptible.

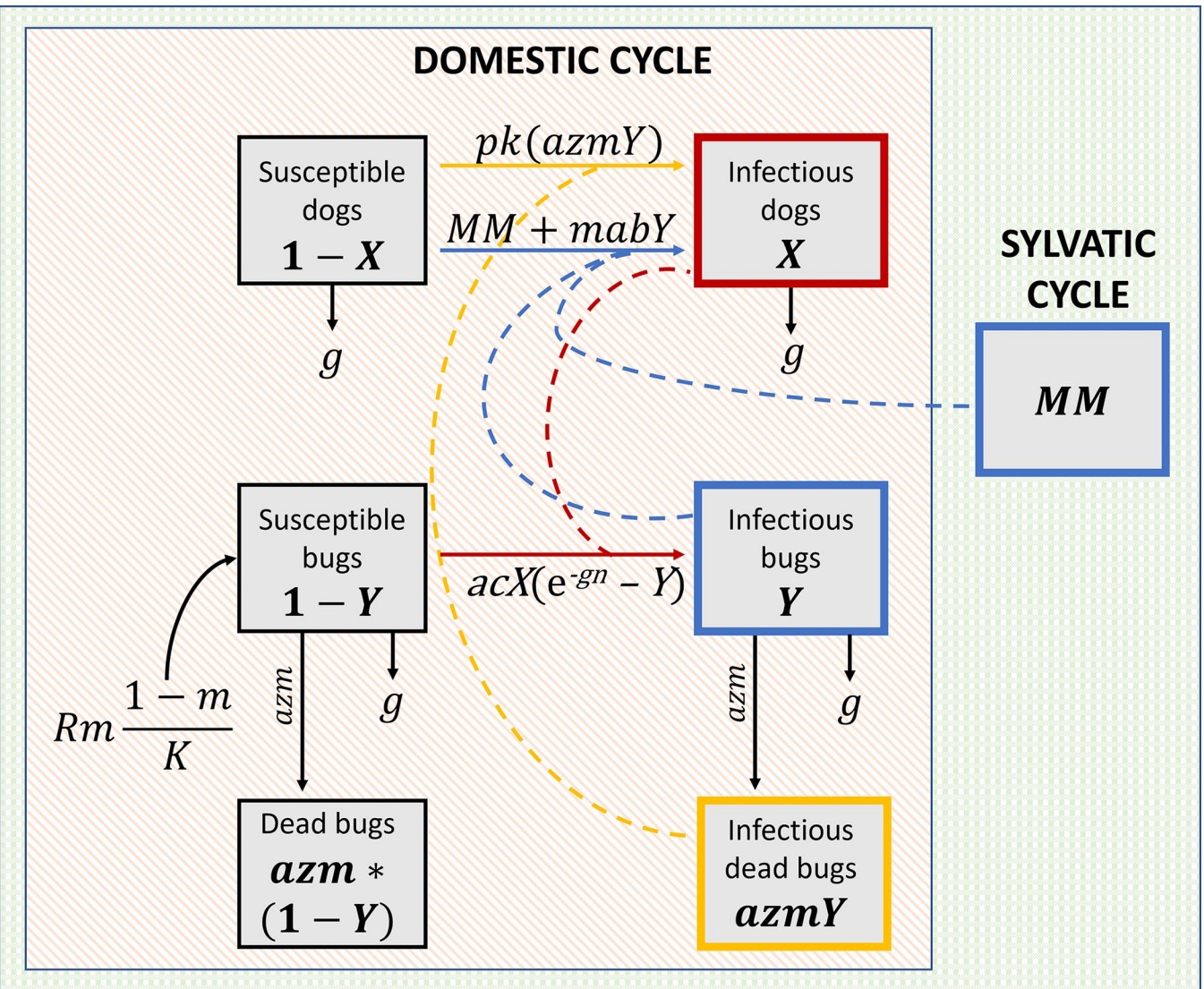

**Fig 5. Mathematical models of *T. cruzi* transmission dynamics between dogs and *T. infestans* in domestic and sylvatic cycles.** Dashed lines represent transmission events and solid lines represent transition between states. Prior to treatment, only vectorial transmission (blue lines) is considered to transition susceptible dogs (*1-X*) to infectious (*X*). Susceptible bugs (*1-X*) are replenished by a logistic birth rate. After administration of fluralaner, there are two transmission routes to infect susceptible dogs: vectorial transmission as before (blue line) and oral transmission (yellow line). In the sylvatic cycle, vectorial transmission is constant due to exposure to external infectious bugs (*MM*).

Dogs move from susceptible to infectious at a rate equal to the force of infection (FOI) due to vectorial transmission, which is equivalent to the product of the bite rate (*a*), probability of transmission from bugs to dogs via biting (*b*), the proportion of infected bugs (*Y*) available, and the ratio of the number of vectors depending on any given host (*m*, ratio of bugs to dogs) in the system. Susceptible and infected dogs can leave the population through the background death rate, *r*; as with prior models, no disease induced mortality is assumed for dogs [10,75]. Susceptible bugs (*1-Y*) become infected (*Y*) at a rate equal to the FOI for vectors, which is the product of the bite rate (*a*), the probability of transmission from dogs to bugs (*c*), and the proportion of infected dogs (*X*); as a susceptible bug must survive the incubation period of *T. cruzi* to become infectious, the FOI also depends the incubation of the parasite within the

vector ($n$) and the daily probability of bug mortality ($g$). As with dogs, susceptible and infected bugs leave the population through the background death rate ($g$). Prior to treatment, transmission dynamics between dogs and bug are represented by the system of differential of equations: $dX$, the change in proportion of infected dogs, $dY$, the change in proportion of infected bugs, and $dm$, the change in the ratio of bugs to dogs in the population (Eqs 1.1, 1.2, 1.3, respectively):

$$dX = mabY(1 - X) - rX \tag{1.1}$$

$$dY = acX(e^{-gn} - Y) - gY \tag{1.2}$$

$$dm = Rm\left(1 - \frac{m}{K}\right) \tag{1.3}$$

Parameter values (Table 1) were adjusted to fit observed prevalence for regions of high and low disease prevalence domestic vectors, and regions with sylvatic vectors. High and low prevalence are used relatively to consider different geographic regions; in our simulations, high prevalence areas have a 2.5 times greater carrying capacity of triatomine per dog than that of low prevalence. Using the model and parameter values in Table 1, the impact of fluralaner treatment on bug and dog transmission dynamics were evaluated over the timescale of decades.

## Treatment model

Data reported from Laiño et al. regarding the percentage of bugs killed after feeding on treated dogs over time were incorporated into the treatment model [33]. We assume that all dogs in a household are treated with fluralaner at a dosage in agreement with manufacturer instructions [52]. The percentage of bugs killed after feeding on treated dogs over days post treatment (DPT) was plotted and fit to a logistic curve (S6 Fig), and the asymptote, x-midpoint and scale values were extracted at timepoints 4–360 DPT. To examine the effects of treatment on different bug populations, the percentage of killed bugs after feeding on treated dogs were taken from data regarding fifth stage pyrethroid-resistant nymphs and fifth stage pyrethroid-susceptible nymphs [33]; analyses in this paper used the data for 5th-stage pyrethroid susceptible nymphs. The values comprising the equation of the logistic curve were incorporated into parameter $z$, the percentage of bugs feeding on treated bugs that are killed at a point in time, and the time dependent covariate was incorporated into the model.

Treatment was initiated into the model after both the bug and dog populations reached equilibrium. To determine these values, the equation for the basic reproductive number of *T. cruzi* was rearranged and solved for $X$ and $Y$, the values of the proportion of infectious dogs and bugs at equilibrium, respectively [61]. Parameter values for regions with semi-sylvatic bugs were calibrated to approximately values reported for *T. dimidiata* reported in Yucatan, Mexico [76]. Incorporating treatment, the differential equations are altered (Eqs 2.1, 2.2, 2.3) to reflect the fact that change in proportion of infected dogs is now subjected to an additional FOI due to ingestion of dead infected bugs (Fig 5).

Contact between dogs and the dead bugs depends on the availability of dead bugs at a given time point; this is the product of the bite rate, $a$, the percentage of bugs that will die after feeding on a treated dogs at that given time point, $z$, the proportion of infected bugs $Y$, and the ratio of bugs to dogs in the population, $m$. The rate that susceptible dogs become infected via oral transmission will depend on the product of $azmY$, the probability of transmission via bug ingestion, $k$, and percentage of dead bugs consumed, $p$. The rate at which the ratio of bugs to

dogs decreases in the population is proportional to the bite rate, and the percentage of bugs killed that feed on treated dogs at a given time point (Eq 2.3), while the population is replenished at the rate of the bug logistic birth rate.

$$dX = [mabY + pk(mazY)](1 - X) - rX \tag{2.1}$$

$$dY = acX(e^{-gn} - Y) - gY - mazY \tag{2.2}$$

$$dm = Rm\left(1 - \frac{m}{K}\right) - maz \tag{2.3}$$

As all bugs in the previous model are assumed to be subjected to fluralaner treatment, the model would not properly represent regions where triatomine vectors include sylvatic bugs. To account for external bugs not affected by treatment, a constant was introduced (*MM*) to the FOI for dogs through vectorial transmission (Eq 3.1). The constant *MM* represents sylvatic infected bugs that can contribute to the vectorial FOI in dogs but would not contribute to the oral FOI if killed by treatment and whose populations would not be reduced if some individuals are killed by fluralaner (Fig 5). Values for the constant were derived by running the model without treatment and determining their impact on infection prevalence in dogs.

$$dX = [mabY + MM + pk(mazY)](1 - X) - rX \tag{3.1}$$

We performed sensitivity analyses upon input parameters based on a range of plausible values found in the literature (S2–S4 Figs). We also created a Shiny web application [74] to allow users to simulate the model in a way that can capture regional variation in multiple parameters available at https://jrokh.shinyapps.io/NewExternalBugs/. All analyses were carried out assuming the dogs consume 80% of the bugs killed by treatment ($p = 0.8$). We also explored different consumption levels from 20% to 80% (S2–S5 Figs). Unless explicitly stated, all models were run using the baseline parameter values (Table 1). R code used within the shiny application and to run the different simulations overviewed here can be found in S1 Code.

## Supporting information

**S1 Fig. Proportion of infected dogs and infected *T. infestans* prior administration of fluralaner treatment.** A corresponds to the baseline pre-treatment model in regions with high disease prevalence and domestic vectors. B corresponds to the baseline pre-treatment model in regions with low prevalence and sylvatic vectors.
(TIF)

**S2 Fig. Sensitivity analysis on proportion of consumed bugs by dogs for single fluralaner treatment in a high prevalence region (corresponds to Fig 1 in main text).** (A) corresponds to 20% of bugs being consumed; (B) corresponds to 50% of bugs being consumed; (C) corresponds to 80% of bugs being consumed.
(TIF)

**S3 Fig. Sensitivity analysis on proportion of consumed bugs by dogs for multiple fluralaner treatments in a high prevalence region (corresponds to Fig 2 in main text).** Annual administration of fluralaner for both 4 years (A) and 6 years (B) was simulated, as well as administration every 90 days (veterinary recommendation) for one year (C) and for two years (D).
(TIF)

**S4 Fig. Sensitivity analysis on proportion of consumed bugs by dogs for fluralaner treatments schemes in a low prevalence region (corresponds to Fig 3 in main text).** We explored

a range of dog average lifespan from 3 years (A-C) to 6 years (D-F). Treatment scenarios include one time treatment (A, D), annual treatment for 4 years (B, E), and treatment every 90 days for 1 year (C, F).
(TIF)

**S5 Fig. Sensitivity analysis on proportion of consumed bugs by dogs for fluralaner treatments schemes in a low prevalence region with semi sylvatic cycles (corresponds to Fig 4 in main text).** We explored a range of dog average lifespan from 3 years (A-C) to 6 years (D-F). Treatment scenarios include one time treatment (A, D), annual treatment for 4 years (B, E), and treatment every 90 days for 1 year (C, F).
(TIF)

**S6 Fig. The percentage of bugs killed after feeding on treated dogs over days post treatment.** Data from Laino et al (2019) on the declining percentage of bugs killed after feeding on fluralaner treated dogs was fit to a logistic curve and incorporated into the model of T. cruzi transmission dynamics in bugs and dogs with fluralaner treatment. Initial analyses used the data from 5th stage pyrethroid susceptible nymphs.
(TIF)

**S1 Code. Output rendered from the R markdown used to run the simulations addressed.**
(HTML)

## Acknowledgments

JLR would like to acknowledge the expertise and guidance of her advisor, Dr Thersa Sweet at Dornsife School of Public Health at Drexel University.

## Author Contributions

**Conceptualization:** Jennifer L. Rokhsar, Michael Z. Levy, Ricardo Castillo-Neyra.

**Formal analysis:** Jennifer L. Rokhsar, Brinkley Raynor, Justin Sheen, Neal D. Goldstein, Michael Z. Levy, Ricardo Castillo-Neyra.

**Funding acquisition:** Ricardo Castillo-Neyra.

**Investigation:** Jennifer L. Rokhsar, Michael Z. Levy, Ricardo Castillo-Neyra.

**Methodology:** Jennifer L. Rokhsar, Brinkley Raynor, Justin Sheen, Michael Z. Levy, Ricardo Castillo-Neyra.

**Project administration:** Michael Z. Levy, Ricardo Castillo-Neyra.

**Resources:** Michael Z. Levy.

**Supervision:** Michael Z. Levy, Ricardo Castillo-Neyra.

**Visualization:** Jennifer L. Rokhsar, Brinkley Raynor.

**Writing – original draft:** Jennifer L. Rokhsar, Brinkley Raynor, Justin Sheen, Michael Z. Levy, Ricardo Castillo-Neyra.

**Writing – review & editing:** Jennifer L. Rokhsar, Brinkley Raynor, Justin Sheen, Neal D. Goldstein, Michael Z. Levy, Ricardo Castillo-Neyra.

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
