## [Decision Letter · Decision Letter 0]

21 Sep 2022

Dear Miss Raynor,

Thank you very much for submitting your manuscript "Modeling the impact of treating dogs with fluralaner to halt Trypanosoma cruzi transmission" for consideration at PLOS Computational Biology.

As with all papers reviewed by the journal, your manuscript was reviewed by members of the editorial board and by several independent reviewers. In light of the reviews (below this email), we would like to invite the resubmission of a significantly-revised version that takes into account the reviewers' comments.

The Authors are expected to address all the criticisms by all Reviewers. In particular, provide more description and assessment of the infectiousness of killed bugs (Reviewer #1 and #3), clarify if the effectiveness of insecticide is well supported and perform some model validation (Reviewer #2), provide more description of whether and how often dogs eating dead insects (Reviewer #3). In additional to the above comments, please address,

1. Discussion “…we assumed that dogs consumed 80% of bugs killed by treatment and that consumption happened immediately upon death. Could the authors assess how realistic are these two assumptions? Is there any direct or indirect evidence to support 80% of killed bugs were consumed?

2. Sensitivity analyses have been carried out for the parameters and was made available in Shiny. However, the main conclusion was based on only one specific set of parameters but there seems to be substantial uncertainty (Table 1). Some of these sensitivity analyses should be presented in the results and also discussed.

3. The main conclusion should take into account the sensitivity analyses, which inform how likely or under what situation there may be potential harm from xenointoxiciation.

We cannot make any decision about publication until we have seen the revised manuscript and your response to the reviewers' comments. Your revised manuscript is also likely to be sent to reviewers for further evaluation.

Sincerely,

Eric HY Lau, Ph.D.

Academic Editor

PLOS Computational Biology

Thomas Leitner

Section Editor

PLOS Computational Biology

The Authors are expected to address all the criticisms by all Reviewers. In particular, provide more description and assessment of the infectiousness of killed bugs (Reviewer #1 and #3), clarify if the effectiveness of insecticide is well supported and perform some model validation (Reviewer #2), provide more description of whether and how often dogs eating dead insects (Reviewer #3). In additional to the above comments, please address,

1. Discussion “…we assumed that dogs consumed 80% of bugs killed by treatment and that consumption happened immediately upon death. Could the authors assess how realistic are these two assumptions? Is there any direct or indirect evidence to support 80% of killed bugs were consumed?

2. Sensitivity analyses have been carried out for the parameters and was made available in Shiny. However, the main conclusion was based on only one specific set of parameters but there seems to be substantial uncertainty (Table 1). Some of these sensitivity analyses should be presented in the results and also discussed.

3. The main conclusion should take into account the sensitivity analyses, which inform how likely or under what situation there may be potential harm from xenointoxiciation.

Reviewer's Responses to Questions

**Comments to the Authors:**

Reviewer #1: Yes

Reviewer #2: The use of insecticide against the dog for killing the vector of Trypanosoma cruzi can be considered to trigger two different impacts on the infection risk with Trypanosoma cruzi, killing the vector can decrease infection risk, however, the dead vector can be consumed and increase infection risk. The estimation of effectiveness of insecticide requires mathematical modelling due to the nature of this conflicting efficacies of insecticide, and the authors constructed the model and simulated with several scenarios. Although this hypothesis regarding the conflicting efficacies of insecticide is interesting, the evidence is poor. At least, the assessment of the effectiveness of insecticide using observed data of infection status of dog with/without insecticide or epidemiological data of dog is required to show how the authors’ hypothesis is realistic and important to characterize Chagas disease among dog population. Also, no model validation with real data is conducted. How this simulation study captures the real Chagas disease well is not shown, I think that this paper is not appropriate for publication in PLoS Computational Biology.

Reviewer #3: Modeling the impact of treating dogs with fluralaner to halt Trypanosoma cruzi transmission.

This is a very interesting manuscript. I believe it is worth publishing after some main points are addressed.

My main comments refer to three points:

1) Rather than stating that authors are modelling effects of treating dogs with fluralaner, I believe that results are ligated to xenointoxication in general. Several compounds besides fluralaner have been assayed for xenointoxication (in ref to Loza et al., 2017) why have the authors narrowed their findings to fluralaner in the title? Indeed the authors state in their abstract that the objective of this work was to: “Examine the potential for increased infection rates of T. cruzi in dogs following xenointoxication”. Thus my suggestion is to change the title to: Modeling the impact of treating dogs with xenointoxication to halt Trypanosoma cruzi transmission. I would also suggest avoiding to particularly refer to fluralaner when xenointoxication in general would also be the case in the rest of the text.

2) Secondly, authors state that dogs can become infected when eating infected triatomines, even when insects are killed by the xenointoxication (i.e.: “infectious dead bugs” in Fig 1). Is there any evidence on the occurrence (and duration) of viability of T. cruzi in dead or moribund insects? Can dead triatomines infect dogs that ingest them? Please provide the corresponding references. Moribund and dead insects are also expected to occur in routine vector control insecticide spraying campaigns, is there any evidence of dogs becoming infected by ingesting insecticide-affected triatomines? Also, although this reader is aware of dogs’ habit of eating live triatomines, flies and other flying insects is there any evidence of dogs eating dead or unmoving insects? In addition, a common scenario share by many endemic T. cruzi areas are chickens moving/roaming freely along the house ecotopes. Chickens indeed eat insects. Which is the expected effect of chickens subtraction by ingestion of dead insects on the reported results ? Is there any reason for not including this process in the model?

3) In ref to lines 335-337. Have the authors explored the results of the models if instead of incorporating a vector birth rate, they allow the vector population to crash? This could readily be the case after insecticide treatment.

Minor comments

Author summary

Lines 62-63. “a more effective transmission pathway than predation of the dogs by the insects.” Please rephrase, T. cruzi is not transmitted by predation.

Lines 90-91 “the probability of transmission due to oral vector ingestion is about 1000 times greater than vectorial transmission”. Is there any empirical data corroborating this mathematical estimation? If not please state so in the introduction. In ref 9 infected bugs were expected to exhibit an altered behavior due to infection increasing exposure to predations. The opposite is expected to occurred when treating with intoxicated triatomines. How have the authors accounted for this?

Line 93. Not only in urban areas.

Lines 96. Please revise extremes, zero seroprevalences have been reported in several countries and also higher prevalences than 60%. I would suggest checking more recent reviews on the topic.

Lines 116-117. Please check the reference provided, there is no mention to vector control cost in the cited manuscript.

Lines 118-119, “treatment of canine reservoirs with insecticide could prove a cost-effective mechanism to reduce T. cruzi infection in people.” No reference has been giving to support this sentence.

I find this statement misleading, it is so vague that what is said can or cannot be true. How many dogs are needed to receive treatment in order to reduce T. cruzi infection in people? Which is the cost of the insecticide employed? with which residual effect? How many rounds of treatments are planned? Please revise.

Table 1. Please check the references provided. Also include the units of the parameters.

Maximum percentage of vectors eaten by dogs in a day. Please provide a reference to support this estimates. Why have the authors not analyzed the sensitivity of the model to changes in this key parameter?

M&M Line 356 and 418-419. I find confusing that it is stated that the “The model considers a single species of host (dogs) and vector (Triatoma infestans) (l.356) ” but in lines 418-419, it is stated “Parameter values for regions with semi-sylvatic bugs were calibrated to approximately values reported in Yucatan, Mexico. There is no T. infestans in Mexico. Please explain/revise.

Discussion

Lines 293-294. “ when dogs consumed a moderate amount of the killed bugs” . I find this expression misleading, 80% of the bugs killed is quite high rather than moderate. Please rewrite this sentence.

References

Loza, A., Talaga, A., Herbas, G., Canaviri, R.J., Cahuasiri, T., Luck, L., Picado, A., 2017. Systemic insecticide treatment of the canine reservoir of Trypanosoma cruzi induces high levels of lethality in Triatoma infestans, a principal vector of Chagas disease. Parasites Vectors 10, 344

**Have the authors made all data and (if applicable) computational code underlying the findings in their manuscript fully available?**

Reviewer #1: Yes

Reviewer #2: None

Reviewer #3: Yes

PLOS authors have the option to publish the peer review history of their article (what does this mean?). If published, this will include your full peer review and any attached files.

Reviewer #1: No

Reviewer #2: No

Reviewer #3: No
---

## [Decision Letter · Decision Letter 1]

4 Jan 2023

Dear Miss Raynor,

Thank you very much for submitting your manuscript "Modeling the impact of xenointoxication in dogs to halt Trypanosoma cruzi transmission" for consideration at PLOS Computational Biology.

As with all papers reviewed by the journal, your manuscript was reviewed by members of the editorial board and by several independent reviewers. In light of the reviews (below this email), we would like to invite the resubmission of a significantly-revised version that takes into account the reviewers' comments.

The Authors have addressed most of the computational issues satisfactorily, however there is a critical issue on biological plausibility where the entire analysis was based on. In particular as pointed out by Reviewer #3, evidence that 1) dogs may eat dead triatomines; 2) dead triatomines may be infectious, are crucial for the validity of the analysis. The manuscript can be further assessed if such evidence could be provided and incorporated in the model.

We cannot make any decision about publication until we have seen the revised manuscript and your response to the reviewers' comments. Your revised manuscript is also likely to be sent to reviewers for further evaluation.

Sincerely,

Eric HY Lau, Ph.D.

Academic Editor

PLOS Computational Biology

Thomas Leitner

Section Editor

PLOS Computational Biology

The Authors have addressed most of the computational issues satisfactorily, however there is a critical issue on biological plausibility where the entire analysis was based on. In particular as pointed out by Reviewer #3, evidence that 1) dogs may eat dead triatomines; 2) dead triatomines may be infectious, are crucial for the validity of the analysis. The manuscript can be further assessed if such evidence could be provided and incorporated in the model.

Reviewer's Responses to Questions

**Comments to the Authors:**

Reviewer #2: My comment has been solved partly and as much as possible at the current stage.

Reviewer #3: My main point regarding this revised version relates to the key assumption that dogs can get infected by eating dead triatomines.

-In reference to the 2nd Response to reviewer 1:

Authors state “In our lab we have infected mice with feces from T. cruzi-carrying triatomines that were 4 days dead“

Very interesting and relevant information! Have the authors published these results?

-In reference to the 4th Response to reviewer 1. In this manuscript the infection of dogs by eating dead triatomines is crucial and is a premise sustaining the whole reasoning and calculations. However, the authors have not provided any evidence that dog can get T. cruzi infection by eating dead triatomines. (Please see my comments on the references provided) Please revise

Ref 27 Gürtler et al., 1986. The only mentioning of oral infection in dogs in this manuscript refers to the potential infection dogs may acquire when eating infected rodents.

Ref 35. Reithinger et al. 2004. The overall disappearance of 12% of exposed bugs to collared and uncollared dogs implies consumption of bugs by the dogs. However, there is no evidence of eating or consumption of dead bugs especially given that the deltametrin-impregnated collars had no effect on triatomine survival.

Ref 50 Bradley et al., 2000. This manuscript mentions that some puppies may have acquired infection by chasing triatomines because they were reported to chase bugs.

It is thus necessary that the authors clearly warn the reader that it is speculative that dogs may become infected by eating dead triatomines (if they prey on dead triatomines at all).

In reference to the Reviewer’s 3 comments:

In reference to my 2nd comment: I believe that authors have provided different references that explain human oral transmission (which is something not questioned at all). The cited references (9,12,15,17) make clear that trypanosomes can survive (and probably multiply) in homemade seasonal fruit juices, however, this is not the kind of oral transmission that could involve dogs (main topic of this manuscript). My point is what evidence exists on the survival of parasite and infectiousness of dead infected triatomines in the household environment. Assuming that parasite survival and infectiousness of dead triatomines are exactly the same in the environment (floor, patio, etc.) than in fruit juices or other food staff is risky. This should be made clear to the reader.

Regarding the consumption of dead insects: Please refer to my comment above on the interpretation of Reithinger et al., 2004 ‘s results.

-I believe that the results obtained when Triatomine populations crash should be included in the revised version (Figure included in the answers to the reviewers), not only as a sentence and authors should discuss on this topic, for example in the 2nd paragraph of the discussion . Please include if this is a share pattern for all transmission scenarios in the final version of the manuscript.

It seems that authors are biased on reporting detrimental effects of xenointoxication and conclude “In regions with low prevalence and domestic or sylvatic vectors, there is potential harm”. However the potential effect of diminishing triatomine population and thus decreasing T. cruzi prevalence both in insects and dogs is neglected. Please revise.

It would have been helpful that line numbers were included in the revised version with tracked changes.

INTRODUCTION

Second paragraph.

“The probability of transmission due to oral vector ingestion is ESTIMATED TO BE about 1000 times greater than vectorial transmission. [16,17] and T. cruzi parasites in feces outside of the bug are viable (infectious) for up to 48 hours IN FRUIT JUICES [17]”

4th paragraph in the revised version authors had added “in particular, fluralaner, a relatively new isoxazoline oral insecticide commonly used to prevent tick and flea infestations, proved especially effective in killing bugs when they fed on dogs under laboratory conditions and is being considered for Chagas control programs [33,36].

Ref 33 concludes: “Fluralaner and xenointoxication are eligible for Phase III efficacy trials alone or combined with other methods in the frame of an integrated vector management strategy in areas with or without pyrethroid resistance”. There is no suggestion on the employment for Chagas control program. Please revise.

**Have the authors made all data and (if applicable) computational code underlying the findings in their manuscript fully available?**

Reviewer #2: None

Reviewer #3: Yes

PLOS authors have the option to publish the peer review history of their article (what does this mean?). If published, this will include your full peer review and any attached files.

Reviewer #2: No

Reviewer #3: No
---

## [Decision Letter · Decision Letter 2]

19 Apr 2023

Dear Miss Raynor,

We are pleased to inform you that your manuscript 'Modeling the impact of xenointoxication in dogs to halt Trypanosoma cruzi transmission' has been provisionally accepted for publication in PLOS Computational Biology.

Best regards,

Eric HY Lau, Ph.D.

Academic Editor

PLOS Computational Biology

Thomas Leitner

Section Editor

PLOS Computational Biology

Reviewer's Responses to Questions

**Comments to the Authors:**

Reviewer #3: My previous comments have been addressed. Thank you!

**Have the authors made all data and (if applicable) computational code underlying the findings in their manuscript fully available?**

Reviewer #3: Yes

PLOS authors have the option to publish the peer review history of their article (what does this mean?). If published, this will include your full peer review and any attached files.

Reviewer #3: No

---

## [Editor Report · Acceptance letter]

2 May 2023

PCOMPBIOL-D-22-00771R2 

Modeling the impact of xenointoxication in dogs to halt *Trypanosoma cruzi* transmission

Dear Dr Castillo-Neyra,

I am pleased to inform you that your manuscript has been formally accepted for publication in PLOS Computational Biology. Your manuscript is now with our production department and you will be notified of the publication date in due course.

With kind regards,

Zsofia Freund
